# A Computer-Assisted System for Early Mortality Risk Prediction in Patients with Traumatic Brain Injury Using Artificial Intelligence Algorithms in Emergency Room Triage

**DOI:** 10.3390/brainsci12050612

**Published:** 2022-05-07

**Authors:** Kuan-Chi Tu, Tee-Tau Eric Nyam, Che-Chuan Wang, Nai-Ching Chen, Kuo-Tai Chen, Chia-Jung Chen, Chung-Feng Liu, Jinn-Rung Kuo

**Affiliations:** 1Department of Neurosurgery, Chi Mei Medical Center, Tainan 710402, Taiwan; gary12223@hotmail.com (K.-C.T.); ronaldowen@gmail.com (T.-T.E.N.); wangchechuan@gmail.com (C.-C.W.); 2Center for General Education, Southern Taiwan University of Science and Technology, Tainan 710402, Taiwan; 3Department of Nursing, Chi Mei Medical Center, Tainan 710402, Taiwan; dreampatty11@gmail.com; 4Department of Emergency, Chi Mei Medical Center, Tainan 710402, Taiwan; 890502@mail.chimei.org.tw; 5Department of Information Systems, Chi Mei Medical Center, Tainan 710402, Taiwan; carolchen@mail.chimei.org.tw; 6Department of Medical Research, Chi Mei Medical Center, Tainan 710402, Taiwan; chungfengliu@gmail.com

**Keywords:** artificial intelligence, machine learning, traumatic brain injury, mortality, emergency room triage, computer-assisted system

## Abstract

Traumatic brain injury (TBI) remains a critical public health challenge. Although studies have found several prognostic factors for TBI, a useful early predictive tool for mortality has yet to be developed in the triage of the emergency room. This study aimed to use machine learning algorithms of artificial intelligence (AI) to develop predictive models for TBI patients in the emergency room triage. We retrospectively enrolled 18,249 adult TBI patients in the electronic medical records of three hospitals of Chi Mei Medical Group from January 2010 to December 2019, and undertook the 12 potentially predictive feature variables for predicting mortality during hospitalization. Six machine learning algorithms including logistical regression (LR) random forest (RF), support vector machines (SVM), LightGBM, XGBoost, and multilayer perceptron (MLP) were used to build the predictive model. The results showed that all six predictive models had high AUC from 0.851 to 0.925. Among these models, the LR-based model was the best model for mortality risk prediction with the highest AUC of 0.925; thus, we integrated the best model into the existed hospital information system for assisting clinical decision-making. These results revealed that the LR-based model was the best model to predict the mortality risk in patients with TBI in the emergency room. Since the developed prediction system can easily obtain the 12 feature variables during the initial triage, it can provide quick and early mortality prediction to clinicians for guiding deciding further treatment as well as helping explain the patient’s condition to family members.

## 1. Introduction

Traumatic brain injury (TBI) remains a well-known public health challenge. The incidence of TBI in Taiwan was 220.6 per 100,000 person-years and the standardized in-hospital mortality rate was 10.7 deaths per 100,000 person-years from 2007 to 2008 [1]. In the USA, statistics have shown that it has been steadily increasing from 521 in 2001 to 824 in 2010 per 100,000 person-years [2]. The prognosis of head injury patients can be predicted based on age [3], sex [4], obesity [5,6], Taiwan Triage and Acuity Scale (TTAS) [7], Glasgow coma scale (GCS) [8], pupillary light reflex [9], degree of midline shift in computed tomography (CT) scan [10], elevated white blood cell count [11], coagulopathy [12], and presence of comorbidities [13,14].

Steyerberg and colleagues demonstrate development prognostic scores based on admission characteristics including CT scan findings and laboratory results [15]. However, a useful mortality prediction model evaluated by combining multiple risk factors without image study and blood sampling analysis for TBI patients in the emergency triage station has not yet been fully established. Therefore, this study believes that developing an accurate method for predicting mortality risk can be truly helpful in clinical practice. Furthermore, critical care physicians could use this information to help them decide which management such as conservative or aggressive treatment to be given to the patient and to help them educate the family members.

Recently, hospitals have begun to apply statistical and artificial intelligence models with a variety of algorithms such as logistic regression (LR) [16], random forest [RF] [17], support vector machines (SVM) [18], LightGBM [19], XGBoost [20], and multilayer perceptron (MLP) [21], in clinical practice for providing diagnosis suggestions [22], identifying adverse events [23] and surgical complications [24], and determining in-hospital mortality [25,26,27]. These results were based on machine learning and fundamental statistical insights combined with a modern high-performance computing system to learn patterns that can be used for recognition and prediction. However, their predictive performances were found to be variable in different situation [26,27,28,29]. Furthermore, there are few studies focusing on using machine learning models to predict the mortality of patients with TBI [30,31,32,33,34], and it remains unseen in the emergency room triage.

Using the big-data-driven approach and machine learning integrated in the Chi Mei Center’s hospital information system (HIS), the hospital administration built a real-time prediction system for patients with chest pain and major adverse cardiac events in the emergency department (ED) [32,33,34,35]. At present, TTAS can precisely determine which treatment should be prioritized to avoid over-triage and deploy more appropriate resources for ED patients [7]. In current study, we wanted to know whether TTAS combined with other feature variables obtained the best performance in predicting in hospital mortality risk in patients with TBI. The purpose of this study was to construct an early in-hospital mortality risk prediction model for TBI patients of various severities upon their arrival at the initial triage setting based on the HIS data using machine learning algorithms. The present research designed a retrospective study to establish a useful outcome prediction system using the hospital’s TBI clinical database, which will provide scientific data that could be used by healthcare practitioners as reference for choosing the appropriate treatment and for educating patients’ family members.

## 2. Materials and Methods

### 2.1. Ethics

This study obtained ethics approval (10911-006) from the institutional review board of the Chi Mei Medical Center, Tainan, Taiwan. The authors carried out all methods in accordance with relevant guidelines and regulations. The Ethics Committee waived the requirement for informed consent due to the retrospective nature of the study.

### 2.2. Flow Chart of Current Study

Our study was in performed in compliance with transparent reporting of a multivariable prediction model for individual prognosis or diagnosis (TRIPOD) standards.

Figure 1 shows the flow chart for integrating the AI prediction model for patients with TBI in the ED. This study selected 12 feature variables for models training and made several models for comparison with fewer features along with TTAS by significant feature (*p* < 0.05) variables between mortality and non-mortality groups and included those with better correlation between feature variables and mortality based on the correlation coefficient matrix [33,34,35,36]. The authors trained the models on 70% training data and executed the validations through a 30% test set created by a random split. After, six models were constructed to predict mortality risk.

Significance testing was performed by *t*-test for numerical variables and Chi-square test for categorical variables. In addition, we performed Spearman correlation analysis to show the strength of correlation between each feature and outcome. Due to an imbalanced outcome class (mortality) in the dataset, we applied the synthetic minority oversampling technique (SMOTE) [34,35,36,37] to oversample the positive outcome cases (mortality) to be equal to the negative ones (survival) for the final model training with each machine learning algorithm.

### 2.3. Patient Selection

This study retrospectively enrolled all TBI patients aged 18 years old and above admitted to the emergency room (ER) from 1 January 2010 to 31 December 2019 in the electronic medical records of three hospitals under the Chi Mei Medical Group including one medical center, one regional hospital, and one district hospital. 

### 2.4. Features Selection and Model Building

Based on the consensus of our study team consisting of multiple neurosurgeon expert physicians, we selected potential clinical variables based on the following criteria: (i) essential to characterize traumatic brain injury, (ii) routinely acquired/measured, and (iii) easy to interpret with a physical meaning. Then, we used univariate filter methods (including continuous variables and categorical variables; a *p*-value of 0.05 or lower was considered as the selection) and a Spearman’s correlation coefficient and experts’ opinions as the final feature decision. The twelve feature variables were patients’ age, gender, body mass index, TTAS, heart rate, body temperature, respiratory rate, GCS, left and right pupil size, and light reflex due to their wide availability in the triage setting. We used six machine learning algorithms including LR, RF, SVM, LightGBM, XGBoost, and MLP to build a model for predicting in-hospital mortality risk. 

We conducted a grid search with 5-fold cross-validation for hyper-parameter tuning (Appendix A) for each algorithm to better evaluate the model performance and thus obtain the optimal model. A default classification threshold value of 0.5 was used to determine the binary outcome. That is, if the result of the predicted probability was equal to or greater than the threshold, we predicted a positive outcome (mortality); otherwise, we predicted a negative outcome (survival). 

### 2.5. Model Performance Measurement and Calibration

The study used the accuracy, sensitivity, specificity, and AUC (area under the receiver operating characteristic curve) as metrics to measure prediction models’ performance, which have long been used as quantitative performance measurement metrics in health care studies [35,36,37,38] as well as in machine learning modeling [36,37,38,39]. Accuracy represents the proportion of true results, either true positive or true negative, in the targeted population. It measures the degree of veracity of a diagnostic test on a condition. Sensitivity represents the proportion of true positives that are correctly identified by a diagnostic test. It means how good the test is at detecting a disease or a disease outcome (i.e., mortality in our model). Specificity represents the proportion of the true negatives correctly identified by a diagnostic test. It means how good the test is at identifying a normal (negative) condition. An ROC curve (receiver operating characteristic curve) is a graph showing the performance of a classification model at all classification thresholds and AUC measures the entire area underneath the ROC curve representing the degree of separability. 

Meanwhile, models must be well calibrated for patient-level use cases because errors in individual predicted probabilities can lead to inappropriate decision-making [37,38,39,40]. Thus, we also performed model calibration for performance comparison. The Platt scaling method of calibration was used in our models. We then performed a comparison of models with and without calibration in Section 3.5. Comparing model calibration for the best models.

## 3. Results

### 3.1. Demographics and Clinical Pictures in Patients with TBI

The present study included 18,249 patients, of which 9908 were males and 8,341 were females. Their average age was 57.85 ± 19.44 (mean ± SD) years. The average GCS upon arrival at the triage was 14.35 ± 1.94. Further, 266 patients died, with a total mortality rate of 1.44% (266/18,249). A total of 12,334 (67.59%) patients had a level 3 to 5 TTAS. Compared with the non-mortality group, the mortality group had a lower body temperature and BMI [40,41]. Except for heart rate and respiratory rate, all other feature variables were significantly different between mortality and non-mortality groups (Table 1). Due to an imbalanced outcome class in the dataset, this study applied the SMOTE [34,35,36,37] for model training.

### 3.2. The Correlation between Feature Variables and Mortality

To quickly select the proper parameters for machine learning, we conducted a correlation analysis (heat map) of mortality and feature variables using a matrix diagram (Figure 2). It was found that the seven leading feature variables correlated to mortality were left and right pupillary light reflex, GCS, TTAS, left pupil size, age, and right pupil size. This matrix also showed that the GCS, right and left pupillary light reflex, and TTAS were negatively correlated with mortality and age, and that heart rate and pupil size were positively correlated with mortality during hospitalization.

### 3.3. The Predictive Model Using the Twelve Feature Variables

When evaluating the models for mortality risk prediction using the 12 feature variables, this study found that the LR-based model had the best predictive performance (AUC = 0.925), followed by SVM (AUC = 0.920), MLP (AUC = 0.893), XGBoost (AUC = 0.871), random forest (AUC = 0.870), and LightGBM (area under the curve (AUC) = 0.851) (Figure 3). Further, the LR-based model had the highest accuracy (0.893) for mortality risk prediction with a sensitivity of 0.812, and specificity of 0.894 (Table 2).

### 3.4. The Predictive Models Using Fewer Feature Variables

In addition to TTAS, we made attempts to build models with fewer other feature variables for prediction power comparison based on the correlation coefficient. One-feature model used TTAS as the single feature; five-feature model used features of TTAS, left and right pupil light reflex, GCS and left pupil size; six-feature model used features of TTAS, left and right pupil light reflex, left pupil size, age, and right pupil size; seven-feature model used features of TTAS, left and right pupil light reflex, GCS, left pupil size, age, and right pupil size. Model performances were reported in order in Table 3. The result reveals that even only used TTAS as the only one feature, the model still has accepted performance (AUC = 0.872). 

We conducted the Delong test [41,42] to judge whether one model had a significantly different AUC than another model. According to the *p*-values in the cells with a 0.05 level, it revealed that the 12-, 7-, and 6-feature models had insignificant differences from each other (Table 4). It implies that if hospitals are unable to prepare complete data of 12 features for patients, they can consider using 7- or 6-feature models and still maintain excellent prediction performance similar to the best 12-feature model.

### 3.5. Comparing Model Calibration for the Best Models

We conducted the model calibration to decrease the error between predicted probabilities and observed probabilities for preventing inappropriate prediction. The result showed calibrated models (Table 5), which were used as the basis for practical implementation, with a slightly higher performance than uncalibrated ones (e.g., for the 12-feature model, AUC: 0.925 change to 0.926; 7-feature model, AUC: 0.909 change to 0.910).

### 3.6. Distribution of the Predictive Value of Mortality in Each Patient 

Of the patients in the testing set, 80 patients died and 5395 patients survived. Using a box plot with median and interquartile ranges, Figure 4a shows the distribution of the predicted probabilities with the 12-feature calibrated model of all survival patients and all dead patients. We used 0.5 as the threshold for judging the predicted result. Equal to or greater than 0.5 would be predicted to be mortality, otherwise as survival. We deleted the patients whose predictions were inaccurate in Figure 4a to be Figure 4b. Table 6 shows that the predictive value of mortality risk was 10.89% (minimal), 58.76% (25th), 91.15% (median), 99.07% (75th), and 99.94% (maximal), and the predictive value of non-mortality risk was 0.09% (minimal), 4.38% (25th), 11.62% (median), 27.55% (75th), and 99.86% (maximal) (Figure 4a). In accurately predicted true patients, the predicted value of mortality risk was 51.24% (minimal), 74.09% (25th), 96.95%, (median) 99.38% (75th), and 99.94% (maximal), and the predictive value of non-mortality risk was 0.09% (minimal), 3.80% (25th), 9.65% (median), 20.52% (75th), and 49.9% (maximal) (Figure 4b).

### 3.7. External Validation and Computer-Assisted System Development 

To confirm the performance of the AI mortality risk prediction model, this study collected 200 new patients with the same definitions of features and outcomes in the HIS from 10 September 2020 to 10 November 2020 for further external validation (Table 7). The result showed the values of sensitivity (100%), specificity (84.3%), accuracy (84.5%), PPV (0.088), and NPV (1.0). It revealed that this study’s model is acceptably stable and reliable for supporting clinical decision-making.

After we confirmed the best LR-based model, we developed a web-based AI prediction system with the best model and integrated it with the existing emergency triage system to assist clinicians and nurses for better decision making and communication with patients and/or their family members (Figure 5).

Table 8 shows a comparison with related studies using machine learning models to predict in-hospital mortality.

## 4. Discussion

This study reviewed related literature and found that this is an innovative study to develop an early mortality risk prediction system using AI algorithms in the emergency triage setting. The result showed that the model of the TTAS together with the 11 feature variables had the best predictive performance (AUC = 0.925). This study showed remarkable results that can be useful in the field of neurocritical care in the ED: (1) Even without imaging studies or laboratory data collection, our twelve feature variables were highly accurate and better than TTAS in predicting mortality risk in the emergency triage setting; (2) the LR-based model had the highest accuracy and the best performance model for in-hospital mortality risk prediction; (3) if the value of the mortality risk calculation result is greater than 91.15%, the emergency physician must pay extra attention in caring for the patient and explain to the family that the patients’ chance for survival is low; (4) this study actually integrated the best model into the existing HIS for clinical use.

Consistent with previous studies [3,4,7,8,9], our results showed that patients who are older, male, with a level 1 to 2 TTAS, have low GCS, without pupillary light reflex, and larger pupil size have significantly higher mortality risk. Furthermore, it was found that those with low body temperature (36.30 ± 0.70 °C) and low BMI (22.68 ± 3.78) have significantly higher mortality risk. 

Correlation coefficient matrix using Spearman rank order correlation method is a good statistic method to analyze the relationship between two items being observed [36,37]. The correlation coefficient can range from −1 to 1, with −1 or 1 indicating a perfect relationship [40]. This study’s results showed that pupillary light reflex, GCS, and TTAS had low correlation to mortality (correlation coefficient 0.1–0.39), and other feature variables had weak correlation to mortality (correlation coefficient < 0.1). This is possibly the case because compared with conventional clinical study, we used all original data without matching the case group and the control group to diminish the effects of confounding factors; therefore, the data was highly diverse. Moreover, the observation end points were obtained at discharge, which may have been affected by other factors present during this period; thus, future studies should develop a dynamic real-time assessment system to enhance the practicality of the model. 

Based on the correlation coefficient matrix (Figure 2), the pupillary light reflex had the highest correlation to mortality. Both the pupillary light reflex and GCS had a higher correlation to mortality than TTAS. This result implies that the TTAS should be modified to suit the needs of patients with TBI in the emergency triage setting for the early prediction of mortality risk. 

The computerized triage system, the Taiwan Triage and Acuity Scale (TTAS), adapted from the Canadian Triage and Acuity Scale (CTAS, 2017) [42,43,44], was officially launched to avoid over-triage and deploy more appropriate resources for ED patients in Taiwan in 2010 [7]. In the current study, it was found that single TTAS has high AUC (0.876) and sensitivity (0.900) but has low specificity (0.693) and accuracy (0.696) for mortality risk prediction. Based on the results, TTAS combined with eleven other feature variables (AUC = 0.925) obtained the best performance in predicting in-hospital mortality risk in patients with TBI (Table 2). Why is our in-hospital predictive mortality risk model higher than TTAS alone? This may be related to the components of the TTAS, including parameters such as respiratory, hemodynamic, temperature, cognitive impairment and trauma mechanisms. In addition to TTAS, we also include age, gender, BMI, pupil size, and pupil light reflex in our predictive models, which are considered prognostic factors for trauma patients. Although there is still much to learn about its benefits, we recommend the twelve feature variables in the AI predictive model to become integrated in the ER triage for clinical applications.

Figure 4a showed the probabilities of the edge points, indicating the median value was 91.15% and that no mortality risk occurred below 10.89%. The same statistics, excluding the false-positive and true-negative cases, are shown in Figure 4b, indicating that the minimal predictive probability of mortality risk was 51.24%. Therefore, we recommend the following: (1) the emergency physician in the triage setting should pay extremely close attention to patients with a predictive value of mortality risk higher than 51.24%, especially those higher than 91.15%; and (2) the treatment protocols should be different among patients with a predictive value of mortality risk higher than 91.25%, between 91.15% and 51.24%, between 51.24% and 10.89%, and below 10.89%.

In this study, despite obtaining a high mortality risk prediction of more than 91.15%, 136 patients survived. We checked their detail data in our emergency system and found that their survival was due to provision of early intervention such as immediate aggressive resuscitation in the ER (3.7%), craniotomy procedure to remove intracranial hemorrhage (38.9%), and early admission to ICU (86.1%). It implied that although the AI prediction model has excellent predictive performance, it can only be regarded as a decision-support tool rather than a diagnostic determinator.

In the subgroup analysis, this study investigated 15 patients with a mortality risk prediction of less than 50% but who did not survive; six patients died from advanced stage cancer-related complications, five patients died due to delayed intractable intracranial hemorrhage, three patients died due to septic shock, and one patient died because of a concurrent malignant middle cerebral artery infarction. This indicates that patients with TBI may be relatively well during their initial assessment; however, underlying disease, delayed hemorrhage, or other complications may aggravate their condition. Therefore, in order to improve the performance of this model, future studies on specific early intervention and advanced cancer-related complications are necessary.

Furthermore, although the accuracy, sensitivity, specificity, and AUC of our proposed AI model were all higher than 0.8, the PPV was low. It could be mainly due to the scarcity of cases with positive outcomes (dead patients) resulting in imbalanced data distribution, and this may cause a very high false-positive rate if deployed. However, for high-risk TBI in the emergency room, the high false-positive rate could still have clinical value but deserves further improvement. This needs to be further explored by follow-up studies.

Table 8 demonstrated a comparison with related studies for predicting in-hospital mortality using machine learning models. Compared to other studies, we have the highest number of cases and the highest predictive power to predict the risk of in-hospital mortality at the time of emergency triage, and the model is currently being used in clinical settings.

Core CRASH [45] and core IMPACT [15] are currently the most common prognostic systems for trauma. CRASH includes age, motor score, pupils, hypoxia, hypotension, brain CT scan, and lab findings (glucose and hemoglobin level). Core IMPACT includes age, GCS, pupils reflecting the light, major extra-cranial injury, and brain CT scan findings. Our prediction system has good results without brain CT and lab components. In the future, the predictive power of three additional prediction systems can be evaluated.

Despite its strengths, this study still has limitations. First, being a retrospective observational study, the feature variables could have been miscoded or biased by many unrecognized confounders which could have affected the mortality of patients with TBI. Second, we did not evaluate other feature variables such as coagulopathy, brain CT scan findings, surgery procedures, and other complications, which could influence the outcome after TBI. Third, as a study of three hospitals of the Chi Mei Medical Group, its results cannot be generalized to other hospitals. Therefore, further external validation is required for more heterogeneous samples to confirm and extend our results. Finally, TTAS as an input feature is a country-specific measurement and may limit the generalizability and adoption of the proposed algorithm outside of Taiwan.

## 5. Conclusions

Without clinical laboratory data and imaging studies, our results showed that the LR algorithm was the best algorithm to predict the mortality risk in patients with TBI in the emergency room triage setting. Since the 12 feature variables during the initial triage can be easily obtained, our developed AI system can provide real-time mortality prediction to clinicians to help them explain the patient’s condition to family members and to guide them in deciding on further treatment. We believe that predicting the adverse outcomes of patients with TBI using machine learning algorithms is a promising research approach to help physicians’ decision-making after patient admission to ER at the earliest possible time.

## Figures and Tables

**Figure 1 brainsci-12-00612-f001:**
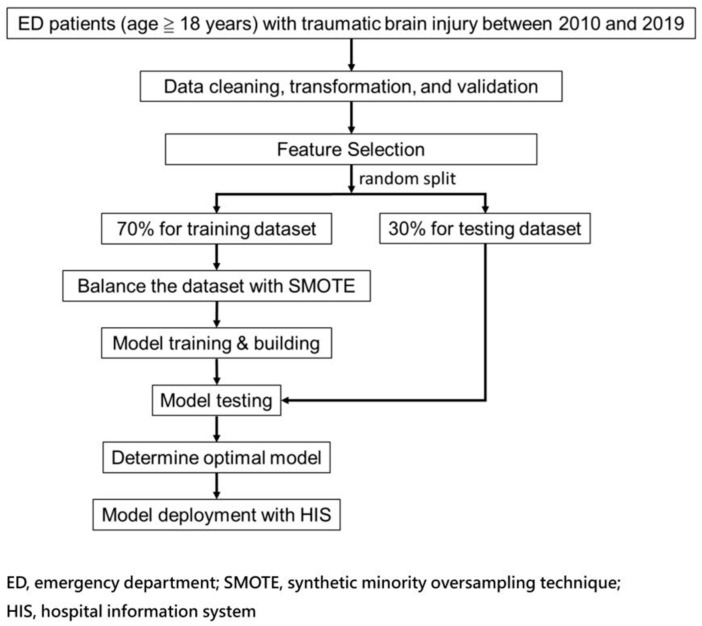
Schematic diagram of the study’s workflow.

**Figure 2 brainsci-12-00612-f002:**
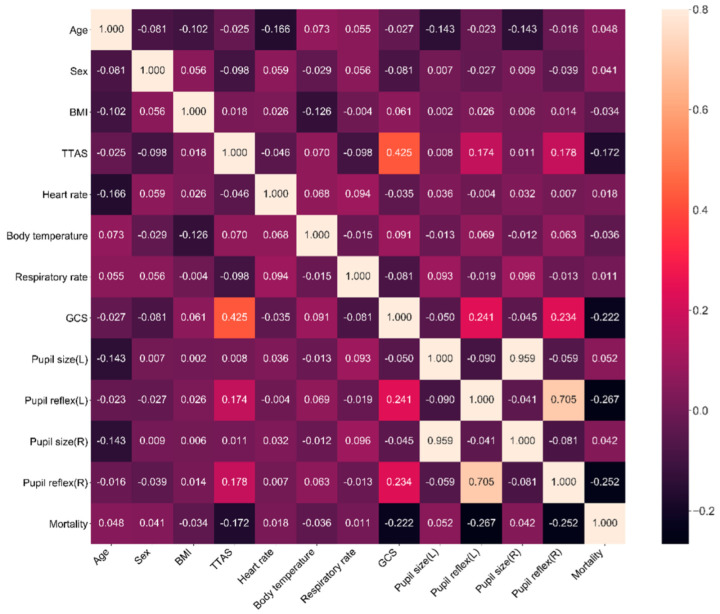
Correlation coefficient matrix using Spearman rank–order correlation for feature variables selection.

**Figure 3 brainsci-12-00612-f003:**
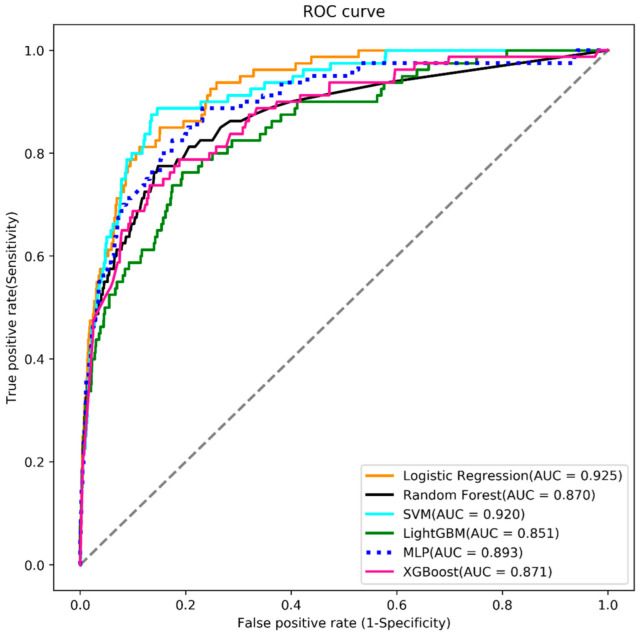
Analysis of receiver operating characteristic curves (ROC), area under the curve (AUC), plotting sensitivity versus 1-specificity for, logistic regression (LR) (orange), random forest (black), support vector machine (SVM) (blue), LightGBM (green), multilayer perceptron (dash), and XGBoost (red) using the 12 feature variables.

**Figure 4 brainsci-12-00612-f004:**
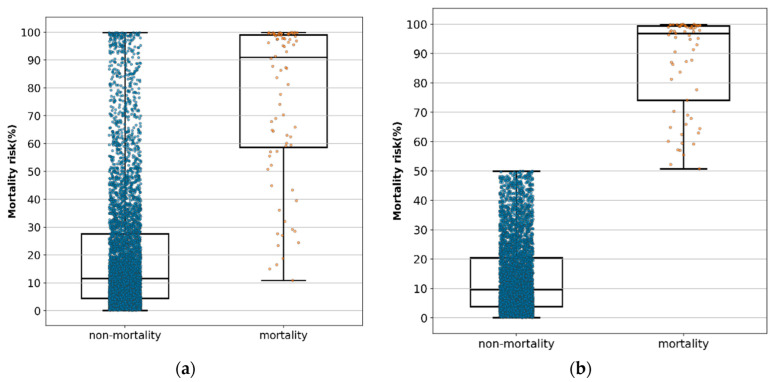
(**a**) Distribution of the predictive value of mortality and non-mortality in each patient by logistic regression model when 12 feature variables were included using a box plot with median and interquartile ranges in all patients. (**b**) Distribution of the predictive value of mortality and non-mortality in each patient by logistic regression model when 12 feature variables were included using a box plot with median and interquartile ranges in all accurately predicted patients.

**Figure 5 brainsci-12-00612-f005:**
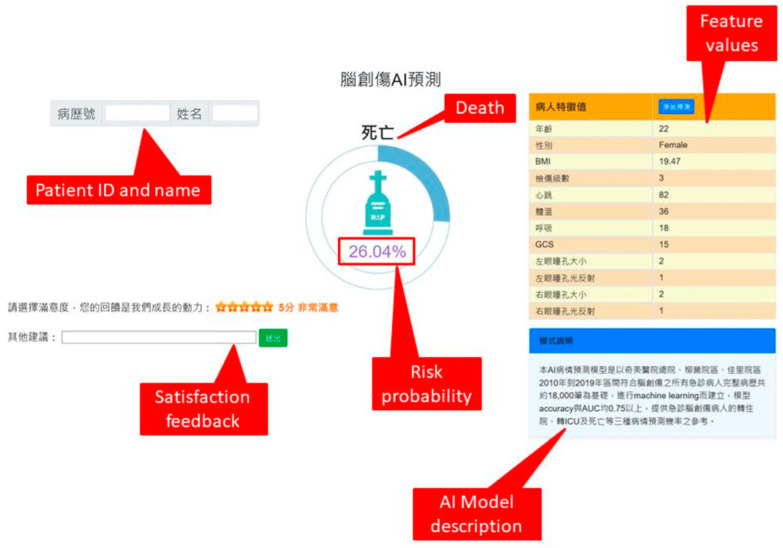
A screenshot of the computer-assisted AI prediction system.

**Table 1 brainsci-12-00612-t001:** Demographics and significances in traumatic brain injury patients.

Variable	Overall n = 18,249	Mortality n = 266	Non-Mortality n = 17,983	*p*-Value
Age, mean (SD)	57.85 (19.44)	65.59 (17.74)	57.73 (19.45)	<0.001
Sex, n (%)				
Female	8341 (45.71)	77 (28.95)	8264 (45.95)	<0.001
Male	9908 (54.29)	189 (71.05)	9719 (54.05)	
BMI, mean (SD)	23.93 (4.45)	22.68 (3.78)	23.95 (4.46)	<0.001
TTAS, n (%)				
Level I	669 (3.67)	147 (55.26)	522 (2.90)	<0.001
Level II	5246 (28.75)	85 (31.95)	5161 (28.70)	
Level III-V	12,334 (67.59)	34 (12.78)	12,300 (68.40)	
Heart rate, mean (SD)	86.59 (18.74)	89.75 (29.66)	86.54 (18.53)	0.080
Body temperature, mean (SD)	36.43 (0.50)	36.30 (0.70)	36.43 (0.49)	0.002
Respiratory rate, mean (SD)	17.65 (2.51)	17.44 (5.12)	17.66 (2.45)	0.502
GCS, mean (SD)	14.35 (1.94)	8.58 (4.68)	14.44 (1.73)	<0.001
Pupil size(L), mean (SD)	2.48 (0.57)	3.05 (1.24)	2.47 (0.55)	<0.001
Pupil reflex (L), n (%)				
−	450 (2.47)	97 (36.47)	353 (1.96)	<0.001
+	17,799 (97.53)	169 (63.53)	17,630 (98.04)	
Pupil size(R), mean (SD)	2.47 (0.57)	2.97 (1.23)	2.47 (0.55)	<0.001
Pupil reflex(R), n (%)				
−	460 (2.52)	93 (34.96)	367 (2.04)	<0.001
+	17,789 (97.48)	173 (65.04)	17,616 (97.96)	

Note: A *t*-test was used for numerical variables and the Chi-square test was used for categorical variables; because there were no cases of mild severity of TTAS levels IV-V in the mortality group, we merged levels III–V for significance testing in demographics.

**Table 2 brainsci-12-00612-t002:** Model performance with 12 features (TTAS + 11 feature variables).

Algorithm	Accuracy	Sensitivity	Specificity	PPV	NPV	AUC (95% CI)
Logistic regression	0.893	0.812	0.894	0.102	0.997	0.925 (0.901–0.950)
Random forest	0.800	0.800	0.800	0.056	0.996	0.870 (0.824–0.916)
SVM	0.865	0.862	0.865	0.087	0.998	0.920 (0.891–0.948)
LightGBM	0.708	0.825	0.706	0.040	0.996	0.851 (0.807–0.895)
MLP	0.825	0.825	0.825	0.065	0.997	0.893 (0.854–0.933)
XGBoost	0.717	0.838	0.715	0.042	0.997	0.871 (0.829–0.914)

Note: PPV = positive predictive value; NPV = negative predictive value; CI = confidence interval; AUC = area under receiver operating characteristic curve.

**Table 3 brainsci-12-00612-t003:** Model performance with fewer feature variables.

Algorithm	Accuracy	Sensitivity	Specificity	PPV	NPV	AUC (95% CI)
(TTAS + 6 feature variables)						
Logistic regression	0.84	0.875	0.839	0.075	0.998	0.909 (0.876–0.943)
Random forest	0.812	0.812	0.812	0.06	0.997	0.885 (0.844–0.925)
SVM	0.806	0.812	0.806	0.059	0.997	0.889 (0.848–0.931)
LightGBM	0.724	0.825	0.722	0.042	0.996	0.884 (0.848–0.920)
MLP	0.808	0.875	0.807	0.063	0.998	0.905 (0.869–0.941)
XGBoost	0.812	0.838	0.812	0.062	0.997	0.897 (0.863–0.931)
(TTAS + 5 feature variables)						
Logistic regression	0.823	0.825	0.823	0.065	0.997	0.907 (0.875–0.939)
Random forest	0.812	0.812	0.812	0.06	0.997	0.876 (0.840–0.913)
SVM	0.824	0.825	0.824	0.063	0.997	0.904 (0.872–0.937)
LightGBM	0.826	0.825	0.826	0.073	0.997	0.883 (0.845–0.921)
MLP	0.814	0.838	0.814	0.062	0.997	0.902 (0.871–0.937)
XGBoost	0.806	0.85	0.806	0.061	0.997	0.887 (0.851–0.923)
(TTAS + 4 feature variables)						
Logistic regression	0.925	0.688	0.928	0.125	0.995	0.891 (0.850–0.931)
Random forest	0.876	0.75	0.878	0.084	0.996	0.855 (0.800–0.911)
SVM	0.81	0.788	0.811	0.058	0.996	0.869 (0.824–0.915)
LightGBM	0.871	0.762	0.873	0.082	0.996	0.866 (0.814–0.917)
MLP	0.868	0.788	0.869	0.082	0.996	0.893 (0.855–0.931)
XGBoost	0.876	0.762	0.877	0.084	0.996	0.866 (0.815–0.918)
(TTAS)						
Logistic regression	0.696	0.9	0.693	0.042	0.998	0.872 (0.832–0.912)
Random forest	0.696	0.9	0.693	0.042	0.998	0.872 (0.832–0.912)
SVM	0.696	0.9	0.693	0.042	0.998	0.872 (0.832–0.912)
LightGBM	0.696	0.9	0.693	0.042	0.998	0.872 (0.832–0.912)
MLP	0.696	0.9	0.693	0.042	0.998	0.869 (0.828–0.911)
XGBoost	0.696	0.9	0.693	0.042	0.998	0.872 (0.832–0.912)

Note. PPV = positive predictive value; NPV = negative predictive value; CI = confidence interval; AUC = Area under receiver operating characteristic curve.

**Table 4 brainsci-12-00612-t004:** DeLong test for the logistic regression models with different features.

	TTAS + 11 Features	TTAS + 6 Features	TTAS + 5 Features	TTAS + 4 Features	TTAS
TTAS + 11 features	1	0.103	0.127	0.043	0.003
TTAS + 6 features	0.103	1	0.777	0.174	0.013
TTAS + 5 features	0.127	0.777	1	0.146	0.001
TTAS + 4 features	0.043	0.174	0.146	1	0.007
TTAS	0.003	0.013	0.001	0.007	1

**Table 5 brainsci-12-00612-t005:** Calibrated model performance (logistic regression).

Model	Accuracy	Sensitivity	Specificity	PPV	NPV	AUC (95% CI)
TTAS + 11 feature variables	0.891	0.812	0.892	0.101	0.997	0.926 (0.901–0.950)
TTAS + 6 feature variables	0.843	0.838	0.843	0.073	0.997	0.910 (0.876–0.943)
TTAS + 5 feature variables	0.822	0.825	0.822	0.064	0.997	0.907 (0.875–0.939)
TTAS + 4 feature variables	0.908	0.713	0.911	0.106	0.995	0.891 (0.851–0.932)
TTAS	0.696	0.900	0.693	0.042	0.998	0.872 (0.832–0.912)

**Table 6 brainsci-12-00612-t006:** The predicted risk probabilities of overall results.

*All Cases without Exclusion*			Excluding False-Positive and False-Negative Cases		
	Non-Mortality	Mortality		Non-Mortality	Mortality
count	5395	80	count	4822	65
mean	20.27	76.02	Mean	13.73	87.15
SD	22.81	27.80	SD	12.40	16.15
min	0.09	10.84	min	0.09	50.77
25%	4.38	58.67	25%	3.79	74.10
50%	11.57	90.96	50%	9.60	96.85
75%	27.61	99.07	75%	20.46	99.41
max	99.86	99.94	max	49.96	99.94

Note: Probability value in percentage (%); SD: standard deviation.

**Table 7 brainsci-12-00612-t007:** Clinical characteristics of 200 patients for external validation.

Variable	Survivaln = 197	Mortalityn = 3	*p*-Value
Gender, n (%)			0.247
male	97 (49.2)	0 (0)	
female	100 (50.8)	3 (100)	
Age, median (IQR)	51 (32–67)	72 (29–95)	0.280
GCS, median (IQR)	15 (15–15)	3 (3–6)	<0.001
Pupil size (L), median (IQR)	2.5 (2.0–2.5)	4.0 (2.0–5.0)	0.137
Pupil size (R), median (IQR)	2.5 (2.0–2.5)	2.5 (2.0–4.0)	0.510
light reflex (L), n (%)			<0.001
−	1 (0.5)	2 (66.7)	
+	196 (99.5)	1 (33.3)	
light reflex (R), n (%)			<0.001
−	1 (0.5)	2 (66.7)	
+	196 (99.5)	1 (33.3)	
TTAS, n (%)			<0.001
Level I	4 (2.0)	3 (100)	
Level II	39 (19.8)	0 (0)	
Levels III–V	154(78.2)	0 (0)	
BMI, median (IQR)	24.6 (22.8–24.6)	19.5 (17.3–19.5)	0.008
BT, median (IQR)	36.4 (36.2–36.7)	36.6 (35.0–37.2)	0.778
HR, median (IQR)	86 (75–97)	98 (86–105)	0.183
RR, median (IQR)	16 (16–18)	18 (10–24)	0.632
predictive value, median (IQR)	28.3 (26.0–35.9)	85.8 (85.7–85.8)	0.003

Note: Continuous variables were reported as the median and interquartile range (IQR). Categorical variables were presented as frequency counts with percentages. Variables were evaluated using Mann–Whitney U test for continuous variables and Fisher’s exact test for categorical variables. *p*-Value of <0.05 was considered to show statistical significance.

**Table 8 brainsci-12-00612-t008:** A comparison with related studies.

Study	This Study	Shi et al., 2013 [32]	Matssuo et al., 2019 [30]	Serviá et al., 2020 [33]
Setting	In the emergency room triage	In-hospital	In-hospital	Intensive care unit
Patient number	18,249	3206	232	9625
Study method	Six ML methods	Two ML methods	Nine ML methods	Nine ML methods
Feature variables	12 feature variables	7 feature variables	11 feature variables	11 variables
Outcome	Mortality	Mortality	Mortality	Mortality
Testing results	0.925	0.896	0.875	0.915
(AUC 95% CI)	(0.901–0.950)	(0.871–0.921)	(0.869–0.882)	(N/A)
Best predicting model	Logistic regression	Artificial neural network	Ridge regression	Bayesian network
Real world implementation	Yes	N/A	N/A	N/A.

ML: machine learning.

## Data Availability

Based on the privacy of patients within the Chi Mei Medical Centers Health Information Network, the primary data underlying this article cannot be shared publicly. However, de-identified data will be shared on reasonable request to the corresponding author.

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
