# Peer review of "A Computer-Assisted System for Early Mortality Risk Prediction in Patients with Traumatic Brain Injury Using Artificial Intelligence Algorithms in Emergency Room Triage"

_brainsci, 2022, doi:10.3390/brainsci12050612_

Round 1

Reviewer 1 Report

Thanks to the authors for clarifying the article.

Reviewer 2 Report

The author responses are satisfactory. Thanks.

This manuscript is a resubmission of an earlier submission. The following is a list of the peer review reports and author responses from that submission.

Round 1

Reviewer 1 Report

The authors present an interesting topic, mortality risk prediction in patients with traumatic brain injury using artificial intelligence algorithms, but this approach has been presented in other international articles. I ask the authors to make a comparison between the results obtained by the article team and the results reported in the literature. Articles reported in the literature also use Artificial Intelligence algorithms (neural networks).
In the recommended comparison, it should be clear what is the differentiating element between their proposal and the algorithms proposed in other articles that solve this issue also through neural networks.
This clarification is important because it will better highlight the proposed solution.

Reviewer 2 Report

Comments

  • there is mention of feature selection is "Based on literature review and clinical experience" , however no further details are provided. This needs to be expanded so that it is better understood how these 12 variables were arrived at. What kind of ML methods were used? filter selection, wrapper? details needed
  • the SMOTE method is applied to address the very imbalanced TBI non-mortality vs mortality cohort. However no details are provided on how this was performed.  Please provide details and numbers on how this was performed and what these numbers look like across folds
  • were the same folds tested across the ML models? what was the similarity across folds if they were not the same?
  • please explain the important of figure 4a and b. it is unclear to me. "Overall patients, 80 patients died and 5,395 patients survived. " what is this referring to across non-mortality and mortality? something doesnt add up numbers wise 
  • it is unclear what the split of non-mortality and mortality is on the 200 new patients that are used for external out of sample validation 

Reviewer 3 Report

The authors present compelling evidence of a successful implementation of machine learning to predict mortality for traumatic brain injury using data from three hospitals. Further, the group provides data for a prospective external validation of the chosen algorithm through an integration with their hospital’s information system. 

I have the following comments:

  1. The Introduction suggests that this is the first attempt at using machine learning (ML) to predict mortality at the time of initial presentation to the emergency room without CT or lab findings. This, however, is not necessarily true and has been described in the following examples:
    1. Warman, Pranav I., et al. "Machine learning for predicting in-hospital mortality after traumatic brain injury in both high-income and low-and middle-income countries." Neurosurgery (2022): 10-1227
    2. Steyerberg, Ewout W., et al. "Predicting outcome after traumatic brain injury: development and international validation of prognostic scores based on admission characteristics." PLoS medicine 5.8 (2008): e165.
    3. MRC Crash Trial Collaborators. "Predicting outcome after traumatic brain injury: practical prognostic models based on large cohort of international patients." Bmj 336.7641 (2008): 425-429.
  2. Of the standards listed, please indicate compliance with Transparent reporting of a multivariable prediction model for individual prognosis or diagnosis (TRIPOD) standards. 
  3. Consider substituting variable names in Figure 2 with more informative names (e.g. instead of “x_iclass” consider “TTAS”).
  4. It is not clear from the methodology at what point in time the mortality of the patient is predicted at. Is it in-hospital mortality, 14-day post discharge, 30-day, etc. ?
  5. Consider listing a full table of hyperparameters explored in the supplemental materials.
  6. The description of the ROC as a “graphic probability curve” does not properly convey the meaning as the ROC does not suggest nor plot probabilities. 
  7. Please define how model calibration was conducted; e.g. Platt scaling or isotonic 
  8. The statistical methodology should be defined in  the methods. e.g. how were confidence intervals constructed, were bootstrap samples needed
  9. An unmentioned limitation is the requirement of TTAS as an input feature is that it is a country specific measurement and may limit the generalizability and adoption of the proposed algorithm outside of Taiwan. 
  10. Why are TTAS Level III - V grouped together in Table 1 whereas Level I and Level II are separated?
  11. While the models with varying features were compared using the DeLong test, it is unclear if the LR model’s improvement over other proposed models was actually statistically significant.
  12. Consider reformatting Figure 3’s title to not include “(y_mortality)”
  13. It is not clear why all of the models, including the single variable model, kept TTAS. Why not keep a variable like pupil reflex which Figure 2 suggests is more highly correlated with the outcome.
  14. In section 3.5 it is not clear what the patient subpopulation represents. If this is only a description of the test set that should be specified. 
  15. The description in section 3.5 of  Figure 4a should be clarified as it is not clear what the patient subsets represent.
  16. Figure 4a and 4b should be plotted for the models after model calibration. Further, consider plotting a calibration curve in lieu of figure 4. 
  17. It is not clear at what threshold (e.g. 0.5, Youden-J, etc.) the statistics of accuracy, sensitivity, specificity, ppv, and npv are calculated at.
  18. It is not clear on what data (i.e. the validation set, or the test set) the model calibration was carried out. Proper calibration should not happen on the test set.
  19. Further, the statistical testing for comparison of the models should happen before not after model calibration.
  20. Please provide summary information of the patients (e.g. GCS distribution) of the patients included in the external validation.
  21. Please provide further clarification how the LR model was confirmed to be “stable and reliable.” 
  22. Please provide commentary on what variables compose TTAS and if there are any interactions with the components and the features included here. 
  23. Please provide commentary on the value of the models in the clinical context proposed here given that their calculated positive predictive value (PPV) are quite low and how the value of the models balances with what will likely be a very high false positive rate if deployed.  
  24. Please offer commentary on the value of the proposed model as opposed to models such as core CRASH and core IMPACT TBI calculators. 
  25. The comment made in line 285 that the patients the model predicted at a high mortality risk were saved due to early intervention is unsupported. If almost all (in the case of early admission to the ICU) or none (in the case of aggressive resuscitation in the ER) of the patient’s in the sub-group received the early intervention, then these interventions are homogenous and the most likely explanation of these patients being misclassified is that the model is not a perfect model not that early intervention was an unexplained confounder. 
  26. The affiliations of the authors need to be re-sorted as the affiliation number 6 comes before the affiliation number 2. 
  27. Finally, as a whole the paper needs further revision for grammar.